# Transcriptome-Based Selection and Validation of Reference Genes for Gene Expression in Goji Fruit Fly (*Neoceratitis asiatica* Becker) under Developmental Stages and Five Abiotic Stresses

**DOI:** 10.3390/ijms24010451

**Published:** 2022-12-27

**Authors:** Hongshuang Wei, Haili Qiao, Sai Liu, Xueqin Yuan, Changqing Xu

**Affiliations:** Institute of Medicinal Plant Development, Chinese Academy of Medical Sciences and Peking Union Medical College, Beijing 100193, China

**Keywords:** *Neoceratitis asiatica*, reference genes, expression stability, qPCR normalization, target gene expression

## Abstract

Goji fruit fly, *Neoceratitis asiatica*, is a major pest on the well-known medicinal plant *Lycium barbarum*. Dissecting molecular mechanisms of infestation and host selection of *N. asiatica* will contribute to the determination of best management practices for pest fly control. Gene expression normalization by Real-time quantitative PCR (qPCR) requires the selection and validation of appropriate reference genes (RGs). Hence, 15 candidate RGs were selected from transcriptome data of *N. asiatica*. Their expression stability was evaluated with five algorithms (∆Ct, Normfinder, GeNorm, BestKeeper, and RefFinder) for sample types differing in the developmental stage, sex, tissue type, and in response to five different abiotic stresses. Our results indicated that the RGs *β-Actin* + *GST* for sex, *RPL32* + *EF1α* for tissue type, *RPS13*+ *EF1α* for developmental stages along with odor stimulation, color induction, and starvation-refeeding stresses, *EF1α* + *GAPDH* under insecticide stress, *RPS13* + *RPS18* under temperature stress, respectively, were selected as the most suitable RGs for qPCR normalization. Overall, *RPS18* and *EF1α* were the two most stable RGs in all conditions, while *RPS15* and *EF1β* were the least stable RGs. The corresponding suitable RGs and one unstable RG were used to normalize a target odorant-binding protein OBP56a gene in male and female antennae, different tissues, and under odor stimulation. The results of *OBP56a* expression were consistent with transcriptome data. Our study is the first research on the most stable RGs selection in *N. asiatica*, which will facilitate further studies on the mechanisms of host selection and insecticide resistance in *N. asiatica*.

## 1. Introduction

*Lycium barbarum* L. (Solanaceae, Lycium) has been used as a traditional Chinese medicinal herb for a period of more than 2000 years [1]. The populations of this herb are mainly distributed in Ningxia Hui Autonomous Region, China (Figure 1A) [2]. The dried ripe fruits of *L. barbarum* (also named wolfberry, goji berry, gouqizi) (Figure 1B) are rich in polysaccharides, carotene, amino acids, vitamins, and trace elements [3]. Studies have proven that consuming goji berries can enhance anti-aging and reduce age-related vision loss, which has been associated with prolonged life [3,4]. In recent years, people’s demand for goji berry is growing every year domestically and internationally, which leads to increasing requirements for high yield and quality [5]. However, the yield and quality of goji berries at harvest are affected by the degree of pest insect infestation during multiple developmental growth stages of *L. barbarum* L. [6,7,8]. A major insect pest of *L. barbarum* is the goji fruit fly, *Neoceratitis asiatica* [8]. Infestation of *L. barbarum* with goji fruit flies can result in fruit-damage rates of up to 80% every year [8]. It is challenging to effectively control this pest in the field because of its hidden behavior and inadequate knowledge of it (Figure 1C,D) [8]. At present, most reports on *N. asiatica* are limited to the studies of biological characteristics and pest control strategies [8,9]. However, the mechanistic understanding of plant infestation and host plant selection by *N. asiatica* is still unclear. Further research on the understanding of the molecular mechanisms behind the behavior of *N. asiatica* will facilitate more effective management of goji fruit flies in the field.

The study of gene expression is a key step in functional analysis and molecular biology research [10]. Real-time quantitative PCR (qPCR) is a commonly used tool to measure the relative mRNA expression levels of target genes [10,11,12]. The qPCR method has been widely used in scientific research, including insect classification and evolution, insecticide resistance, and insect-host-plant interaction [13,14,15]. When qPCR data are analyzed by the 2^−∆∆Ct^ method [16], the most suitable reference genes (RGs) must be selected for normalizing target gene expression to improve both the accuracy and reliability of quantitative results [10,11,12,17]. It is generally believed that ideal RGs should be stably expressed in uniform levels under various experimental conditions. In fact, however, studies indicated that there are no RGs that can be stably expressed under all conditions [12,17]. For example, actin commonly used as an RG in insects does not exhibit equally stable expression for the two closely related species *Helicoverpa armigera* and *Helicoverpa assulta* under the same experimental conditions [18,19]. In addition, the most stable RGs for the same insect species (such as *Bradysia odoriphaga*) are different under distinct insecticide stresses. For example, Elongation factor 1α (*EF1α*), Actin (*actin*), and Ribosomal protein L10 (*RPL10*) were stably expressed after imidacloprid, chlorfluazuron, and phoxim treatments, respectively [20]. Therefore, the most suitable RGs for each experimental condition should be screened to make sure the accuracy of target gene expression in qPCR analysis.

Substantial research has been conducted to evaluate and verify reference genes from different insect species [17,20,21]. The most appropriate RGs have been studied and selected in crop insect pests such as locusts, cotton bollworms, and rice stem borers for mechanistic studies of feeding, oviposition, reproduction, and avoidance of natural enemies [22,23,24]. However, relatively few studies have been conducted on suitable RGs screening for medicinal plant insects [25]. Up to now, there is no report on internal RGs screening related to *N. asiatica*. 

In this study, 15 candidate RGs widely used for qPCR normalization in insects [17,20] were identified from the transcriptomic data of *N. asiatica* adults. Then the expression patterns of these genes were measured by qPCR at multiple developmental stages, for both sexes, for different tissues, and under five different abiotic stresses (odor stimulation, color induction, insecticide treatment, starvation-refeeding, different temperature). The expression stability of those candidate RGs was evaluated using five different algorithms (∆Ct, NormFinder, BestKeeper, GeNorm, and RefFinder). The most stable and optimal combination of RGs was determined by GeNorm and RefFinder analyses. A target gene, odorant-binding protein 56a (OBP56a) involved in odor recognition [26], was used to further verify the most stable or unstable RGs in both sex’s antennae and different tissues and under odor stimulation by qPCR analysis. The proposed research provides the most stable reference genes of *N. asiatica* in different experimental conditions for further studies on the mechanisms of host selection and insecticide resistance in *N. asiatica.*

## 2. Results

### 2.1. Verification of Primer Specificity and PCR Amplification Efficiencies

Fifteen candidate RGs (*α-TUB*, *β-Actin*, *EF1α*, *EF1β*, *GAPDH*, *G6PDH*, *UBC*, *AK*, *GST*, *SDHA*, *TBP*, *RPL32*, *RPS13*, *RPS15*, and *RPS18*) and one target gene (*OBP56a*) of *N. asiatica* were selected for qPCR normalization (Table 1). Sequence details and best BLAST hits for these candidate genes are presented in Appendix A. The specific amplification of each primer pair of candidate RGs (Table 1) was confirmed with qPCR. The specificity of each RG primer was validated by melting curve analysis. As shown in Figure 2, the melting curve of all primer sets exhibited a single amplification peak (Figure 2). The size of amplicons ranged from 92 to 198 bp. The amplification efficiencies (E) for these genes varied from 95.1% for *EF1β* to 104.2% for *RPL32*, and the correlation coefficients (*R*^2^) varied from 0.994 (*RPL32*) to 1.000 (*TBP*) (Appendix A and Table 1).

### 2.2. Expression Profiles of Candidate Reference Genes in N. asiatica

The raw cycle threshold (Ct) values of the 15 candidate RGs for qPCR under eight different experimental conditions were collected and are shown in Figure 3 and Appendix A. The Ct values displayed a wide range from 14.51 (*AK*) to 31.38 (*RPS15*) in all samples, and the average Ct ranged from 16.51 ± 0.06 (*GAPDH*) to 25.57 ± 0.25 (*TBP*), indicating that *GAPDH* had the highest expression level, whereas *TBP* had the lowest expression level in the samples tested. The mean Ct values of *EF1α* (17.12 ± 0.01) in each or total samples with the minimum standard error (SE), indicating that this gene was the most stable of the genes tested under different experimental conditions. Additionally, the most unstable gene was *TBP* in all samples (25.57 ± 0.25) (Figure 3 and Appendix A). The results also showed that *EF1α* exhibited the smallest variation in expression levels, whereas *TBP* displayed the largest variation among all samples.

### 2.3. Stability of Candidate Reference Genes under Eight Experimental Conditions

Firstly, the expression stability of 15 candidate RGs was evaluated by four algorithms (∆Ct method [27], GeNorm [28], NormFinder [29], and BestKeeper [30]), respectively. Then the RefFinder program [31] combined the results of the above four algorithms to calculate their geometric mean to obtain a comprehensive reference gene ranking. The optimal number for normalization of qPCR analysis was determined by GeNorm results (Pairwise value Vn/(n + 1) < 0.15) [20]. The optimal combination RGs were selected by GeNorm and RefFinder results [20,31].

#### 2.3.1. Odor Stimulation

Based on the ∆Ct and NormFinder analyses, *RPS13* and *RPL32* were the most stable genes, while *EF1α* and *RPS18* were by GeNorm and BestKeeper analyses (Table 2 and Table 3). All four algorithms showed that *RPS15* and *TBP* were the least stable genes. GeNorm analysis presented the pairwise value V2/3 as less than 0.15, indicating that the two RGs were the optimal number for the normalization of qPCR data (Figure 4). Comprehensive ranking order for gene stability using RefFinder was as follows (from the most to least stable): *RPS13*, *EF1α*, *RPS18*, *RPL32*, *GST*, *GAPDH*, *β-Actin*, *SDHA*, *α-TUB*, *G6PDH*, *UBC*, *AK*, *EF1β*, *TBP*, *RPS15* (Figure 5 and Appendix A). Therefore, a combination of *RPS13* and *EF1α* was the optimal RGs for the normalization of target gene expression under odor stimulation in *N. asiatica* (Table 4).

#### 2.3.2. Color Induction

The results for color induction indicated that *EF1α* and *RPS13* were the most stable genes, on the contrary, *TBP* and *RPS15* were the least stable genes (Table 2 and Table 3 and Figure 4). The genes Geomean of ranking values analyzed by RefFinder were listed as follows (from the most to least stable): 1.19 (*EF1α*), 1.57 (*RPS13*), 3.13 (*UBC*), 3.66 (*GAPDH*), 5.18 (*AK*), 5.92 *(RPS18*), 6.70 (*GST*), 7.74 (*EF1β*), 9.24 (*β-Actin*), 10.24 (*α-TUB*), 10.46 (*RPL32*), 12.00 (*SDHA*), 13.00 (*G6PDH*), 14.00 (*RPS15*), 15.00 (*TBP*) (Figure 5 and Appendix A). As shown in Figure 3, all pairwise values were less than 0.15 by GeNorm analysis (Figure 4). Combined with the analysis results of RefFinder and GeNorm, the optimal combination RGs were *EF1α* and *RPS13* for normalization of qPCR target gene expression under color induction in *N. asiatica* (Figure 4 and Figure 5 and Table 4).

#### 2.3.3. Insecticide Treatment

The results of ∆Ct showed that *EF1α* and *GST* were the most stable genes, however, *EF1α* and *GAPDH* were with the other three algorithms (GeNorm, NormFinder, and BestKeeper) (Table 2 and Table 3). The *TBP* gene was the most unstable gene with all analysis methods used (Table 2 and Table 3). The RefFinder results indicated the rank order for gene stability was as follows (from the most to least stable): *EF1α*, *GAPDH*, *GST*, *RPS18*, *SDHA*, *RPS13*, *RPL32*, *RPS15*, *α-TUB*, *AK*, *β-Actin*, *G6PDH*, *EF1β*, *UBC*, *TBP* (Figure 5). Among them, *EF1α* and *GAPDH* were considered as the optimal combination RGs for qPCR normalization under insecticide (imidacloprid) stress in *N. asiatica*, as determined from GeNorm results (V2/3 < 0.15) (Figure 4 and Figure 5 and Table 4).

#### 2.3.4. Starvation-Refeeding

The *EF1α* was the most stable gene in all analysis methods used except for the NormFinder method which showed the most stable gene to be *RPS18* (Table 2 and Table 3). All analyses showed that *RPS15* and *EF1β* were the least stable genes (Table 2 and Table 3). Based on the RefFinder, the comprehensive bank order of gene stability was as follows (from the most to least stable): *EF1α*, *RPS13*, *RPS18*, *GAPDH*, *RPL32*, *TBP*, *α-TUB*, *G6PDH*, *GST*, *AK*, *SDHA*, *β-Actin*, *UBC*, *EF1β*, *RPS15* (Figure 5). GeNorm results (V2/3 < 0.15) determined that a combination of two RGs (*EF1α* and *RPS13*) was a better choice for qPCR normalization under starvation-refeeding treatment in *N. asiatica* (Figure 4 and Figure 5 and Table 4).

#### 2.3.5. Different Temperature Treatments

*EF1α* and *RPS18* were the most stable gene in the GeNorm and BestKeeper analyses, whereas *RPS13* in ∆Ct analysis and *AK* in NormFinder analysis (Table 2 and Table 3). Four analysis results indicated the least stable genes were *TBP* and *G6PDH* (Table 2 and Table 3). The RefFinder results revealed that *RPS13* (2.63) and *RPS18* (2.63) had a lower geometric mean and were the most stable genes (Figure 5 and Appendix A). Based on GeNorm analysis, all pairwise values were less than 0.15, except for V14/15 (Figure 4). Thus, *RPS13* and *RPS18* were the optimal combinations for qPCR normalization under different temperature treatments in *N. asiatica* (Figure 4 and Figure 5 and Table 4).

#### 2.3.6. Different Developmental Stages

GAPDH and *RPS13* were the most stable genes in the ∆Ct and NormFinder analysis, while *EF1α* was the most stable gene in the GeNorm and BestKeeper analyses (Table 2 and Table 3). All analyses showed that *TBP* and *G6PDH* were the least stable genes (Table 2 and Table 3). The bank order of gene stability by RefFinder was from the most to least stable genes as follows: *RPS13*, *EF1α*, *GAPDH*, *RPS18*, *GST*, *UBC*, *EF1β*, *RPL32*, *β-Actin*, *SDHA*, *RPS15*, *α-TUB*, *AK*, *TBP*, *G6PDH* (Figure 5). In view of the GeNorm results (Vn/(n + 1) < 0.15), *RPS13* and *EF1α* were proposed as the optimal combination for qPCR normalization in developmental stages of *N. asiatica* (Figure 4 and Figure 5 and Table 4).

#### 2.3.7. Both Sexes

The GeNorm and NormFinder results showed that *β-Actin* and *EF1α* were the most stable genes, whereas the other two analyses had the same gene found to be the most stable *RPS13* (Table 2 and Table 3). TBP was identified as the most unstable gene in all analyses (Table 2 and Table 3 and Figure 5). From the results of RefFinder (from the most to least stable genes): *β-Actin*, *GST*, *RPS13*, *EF1α*, *RPL32*, *SDHA*, *G6PDH*, *UBC*, *GAPDH*, *EF1β*, *AK*, *RPS18*, *RPS15*, *α-TUB*, *TBP*) and from GeNorm analysis (Vn/(n + 1) < 0.15), the optimal combination RGs for qPCR normalization in the male and female of *N. asiatica* adults were *β-Actin* and *GST* (Figure 4 and Figure 5 and Table 4).

#### 2.3.8. Different Tissues

The ∆Ct and NormFinder data indicated *RPL32* and *GST* were the most stable genes. However, GeNorm and BestKeeper results showed that *EF1α* and *RPS18* were the most stable genes (Table 2 and Table 3). All analyses showed that EF1β was the least stable gene (Table 2 and Table 3). The rank order for gene stability determined by RefFinder was as follows (from the most to least stable genes): *RPL32*, *EF1α*, *GST*, *RPS18*, *RPS15*, *GAPDH*, *UBC*, *AK*, *β-Actin*, *SDHA*, *RPS13*, *TBP*, *α-TUB*, *G6PDH*, *EF1β* (Figure 5). Hence, *RPL32* and *EF1α* were considered as the optimal combination RGs for qPCR normalization in different tissues of *N. asiatica*, as determined from GeNorm results (V2/3 < 0.15) (Figure 4 and Table 4).

### 2.4. Comprehensive Ranking Analysis of Candidate Reference Genes

The *EF1α* was the most stable gene in total samples by GeNorm and BestKeeper analyses, while *RPS18* and *GST* were the most stable genes in ∆Ct and NormFinder analyses, respectively (Table 2 and Table 3). The *RPS15* and *EF1β* were the least stable genes in all analyses (Table 2 and Table 3). The overall rank order for gene stability determined from RefFinder results was as follows (most to least stable): *RPS18*, *EF1α*, *GAPDH*, *RPS13*, *GST*, *β-Actin*, *UBC*, *RPL32*, *SDHA*, *AK*, *α-TUB*, *G6PDH*, *TBP*, *EF1β*, *RPS15* (Figure 5). The GeNorm analysis showed that all pairwise variation values were less than 0.15, except for V13/14 and V14/15 (Figure 4). Based on the RefFinder data, *RPS18* and *EF1α* are the most suitable internal RGs for normalizing target gene expression in *N. asiatica* (Figure 4 and Figure 5 and Table 4).

### 2.5. Validation of the Selected Candidate Reference Genes

OBPs can recognize and transport odor molecules to facilitate insects in host plants’ locations [32]. In this study, *OBP56a* of *N. asiatica* was selected as the target gene to verify the expression stability of potential RGs by qPCR at both sexes’ antennae, different tissues, and under odor stimulation. The sequence analysis and amplification efficiencies of *OBP56a* by qPCR were listed in Appendix A. According to our results (Table 4), a single most stable RG or optimal combination RGs and the corresponding most unstable RG were provided for normalizing *OBP56a* expression in *N. asiatica* (Figure 6 and Figure 7). 

As shown in Figure 6, the transcript levels of *OBP56a* in the antennae (male and female insects) and different tissues (antennae, legs, and ovipositor of female) of *N. asiatica* adults were firstly shown by FPKM values of transcriptome data (Figure 6A,B). Further verification by qPCR indicated that target *OBP56a* had a similar expression level by using either single RG or in combination RGs (*β-Actin* + *GST* in sex, *RPL32* + *EF1α* in tissues) to normalize its expression in both sexes and different tissues, respectively. However, the expression profiles of *OBP56a* differed significantly when using the least stable RG (*TBP* or *EF1β*) for normalization (*p* < 0.05) (Figure 6C,D). By comparison, it was found that the expression patterns (mainly in female antenna) of *OBP56a* by using the most stable RGs (*β-Actin* + *GST*, *RPL32* + *EF1α*) for normalization were consistent with the expression trend of transcriptome data in male and female antennae and among different tissues (FPKM) (Figure 6C,D).

Compared to the unstimulated control, the expression of *OBP56a* based on FPKM values were upregulated in the antennae, legs, and ovipositor of female after odor stimulation (host volatiles), especially in the ovipositor (Figure 7A). Meanwhile, the qPCR results for single stable RG (*RPS13* or *EF1α*) or a combination stable RGs (*RPS13* + *EF1α*) normalization showed a similar upregulation trend after odor stimulation (Figure 7B–D). Nevertheless, the *OBP56a* expression was significantly downregulated in the ovipositor under the least stable RG (*RPS15*) normalization (*p* < 0.01) (Figure 7E). After odor stimulation, relative expression levels of *OBP56a* in antennae, legs, and ovipositor of females had a significant difference between the most stable RGs and the least stable RG normalization (*p* < 0.05) (Figure 7F).

## 3. Discussion

*L. barbarum* is a traditional medicinal herb that has been used as a powerful anti-aging agent due to its health benefits [1,4]. Infestation of goji fruit fly *N. asiatica* not only causes the decline of *L. barbarum*’s production and quality but also restricts the healthy development of the *L. barbarum* industry [8]. However, the mechanisms underlying the host selection of *N. asiatica* are still unclear. The method of qPCR is a rapid and accurate method for detecting and analyzing expression levels for genes of interest [11]. RG normalization is a key step needed for enhancing the accuracy of expression level analysis of target genes by qPCR [12]. So far, no study on reference gene screening has been reported in *N. asiatica*. Therefore, we identified 15 commonly used candidate RGs from the transcriptome data of *N. asiatica*. Their expression stability was evaluated by different algorithms to clarify the optimal RGs in eight different experimental conditions for further studies of gene function.

To sum up, our study demonstrated that the expression stability of RGs differed under different experimental treatments and no RG has stable expression in all conditions in *N. asiatica*. This result is in line with previously reported results [10,11,12]. *Actin* was commonly selected as a stable RG for gene expression analysis in developmental stages, sex, tissues, and other different conditions in insects, especially in lepidopteran species [33,34]. However, the current study indicated that *Actin* was only expressed stably in both sexes rather than across developmental stages (*RPS13*) and tissue types (*RPL32*) of *N. asiatica*. In addition, studies on two closely related species of *N. asiatica* showed that three RGs (*TUB*, *GAPDH*, *GST*) for *Bactrocera minax* and *RPL60* for *B. cucurbitae* were the most stable RGs for qPCR analysis under tissue-development condition, however, *actin* was shown to be the most unstable RG [35,36]. It is a normal phenomenon that there are differences in the selection of appropriate RGs under the same conditions among diverse insect species [20,35,36]. Sometimes RGs differ greatly in different developmental stages even in the same tissue [20,36]. Two or more RGs, in fact, can improve the accuracy of qPCR results compared to a single RG [20]. In our study, GeNorm data showed that the pairwise variation value at V2/3 below the proposed 0.15 cut-off value, which indicated that the combination of two stable RGs would be a better choice for qPCR normalization, rather than the use of a single RG under eight experimental conditions. 

In the present study, the stability of RGs in *N. asiatica* could differ under various experimental conditions. Our results were similar to previous studies [17,20,21]. *RPS18* and *EF1α* were the most stable RGs under sex, tissues, developmental stages, and five abiotic stresses in *N. asiatica*. Among them, *RPS13* and *EF1α* were the suitable combination of RGs under developmental stages, odor stimulation, color induction, and starvation-refeeding stresses. *β-Actin* + *GST* and *RPL32* + *EF1α* were stably expressed in both sexes and in the different tissues of *N. asiatica*, respectively. Moreover, *EF1α* + *GAPDH* and *RPS13* + *RPS18* were selected as the optimal combination RGs to normalizing gene expression under insecticide treatment and different temperature stresses, respectively. Previous research has shown that the expression stability of RGs is different under species, growth stage, and biotic and abiotic stresses [17,20,35]. In the larva and adult stages of *B. cucurbitae*, *RPL32* and *RPS13* were the most stable RGs under different temperature stress [37]. In addition, *α-TUB* and *RPL32* were found to be suitable RGs for normalizing the qPCR data in *Phenacoccus solenopsis* under temperature stress [38]. *Actin2* and *α-TUB* were the most appropriate RGs in both males and females of *B. dorsalis*, while *Actin5* + *α-TUB* and *Actin3* + *α-TUB* were considered the best RG combinations for females and males, respectively [39]. Meanwhile, in *Anastatus japonicus*, *Actin* and *EF1α* were optimal RGs in male and female adults [40]. There are a few differences between our results and previous reports for both sexes and under temperature stress [37,39]. When insects were treated with different insecticides, their reference genes could not always express stably [20,41]. For instance, *EF1α* was the most stable RG in *Bradysia odoriphaga* under imidacloprid treatment, while *Actin* was expressed most stably for chlorpyrifos and chlorfluazuron treatments [20]. Furthermore, a previous study also indicated *RPS15* was found to be the most stable reference gene in *Br. odoriphaga* under different developmental stages and different temperature conditions [41]. This result, however, was the opposite of the finding in our study, that *RPS15* was the least stable RG in *N. asiatica*. These findings further confirmed that screening and verification one of the expression stability for RGs, are very important for gene research under each specific experimental condition. 

Randomly selected or unvalidated reference genes are not recommended for direct qPCR normalization of target gene expression, because they cannot ensure accuracy relative to quantification [20,36]. For example, in this study, the expression level of target *OBP56a* in the ovipositor of female *N. asiatica* after stimulation of host plant volatiles was significantly up-regulated using *EF1α* and *RPS13* (the most stable genes) as internal control, while was significantly down-regulated using *RPS15* (the least stable gene) as a reference gene (Figure 7). Validated stable reference genes for normalizing gene expression could obtain better quantitative results [35,36,37,38,39,40,41]. It has also been documented that at least two or more reference genes should be selected for gene quantitative normalization [10]. In this study, OBP56a mainly involved in odor recognition with a conserved six cysteine residues pattern [26,32] was used for reference gene verification (Appendix A). In *N. asiatica*, *β-Actin* and *GST* were the best choices for normalizing *OBP56a* expression in males and females via GeNorm (V2/3 < 0.15) and RefFinder algorithms. qPCR results showed that *OBP56a* had a higher expression in female antennae than in male antennae, which was consistent not only with the transcriptome data of *N. asiatica*, but also with a previous study that used *RPL18*, *RPS17*, and *EF1α* as RGs in *Anastrepha obliqua* [42]. However, using the unstable *TBP* and *RPS15* as RGs failed to detect a significant effect on *OBP56a* expression under both sexual antennae and odor stimulation. Therefore, optimization of reference genes is crucial for the accurate normalization of gene expression, especially for the identification of the relatively subtle difference. To improve the accuracy of target gene quantification, the appropriate reference genes must be screened and verified under the experimental conditions before qPCR detection.

In conclusion, this research is the first to validate reference genes and detect the expression profiles of *OBP56a* in both male and female antennae, different tissues, and odor stimulation in *N. asiatica*. We concluded the corresponding optimal reference genes under eight different experimental conditions based on five algorithms (Table 4). Our findings provide a scale for quantitative gene expression analysis by qPCR under various developmental stages and abiotic stresses in the pest species *N. asiatica*. The purpose of the research is to lay a foundation for future elucidation of the mechanisms for host selection and insecticide resistance in *N. asiatica*. Additionally, our results further emphasize that the most suitable reference genes should be screened and validated under different experimental conditions to ensure the accuracy of gene quantification.

## 4. Materials and Methods

### 4.1. Insect Rearing

*N. asiatica’s* larvae and pupae collected from the main production area of wolfberry in Ningxia, China, were maintained at the Institute of Medicinal Plant Development, Chinese Academy of Medical Sciences, and Peking Union Medical College. After emergence, adults were fed with 10% honey water and reared in a cage (50 cm × 50 cm × 50 cm) made of gauze. The female adults after mating were transferred to a new cage with wolfberry young fruits for further oviposition. The conditions were controlled by an artificial climate cabinet (PXZ-430B, Ningbo Jiangnan Instrument Factory, Ningbo, China) with a 14 Light (L):10 Dark (D) photoperiod at 25 ± 2 °C and 40% relative humidity. 

### 4.2. Collection of Samples under Different Experimental Conditions

The *N. asiatica* adults (males and females) from three days after emergence were used in the experiments. Each experiment was set as three biological replicates. The collected methods used in this study were similar to previous studies [20,33]. The following is detailed information on the sample collection under different experimental conditions. (1) Odor stimulation. The 20 *N. asiatica* adults were exposed to three odor compounds (1 mM methyl salicylate, 1 mM methyleugenol, and 1 mM 4-(p-Acetoxyphenyl)-2-butanone) for 2 h (h) respectively. As a control, the adults were unstimulated with odor compounds. The stimulated and unstimulated adults were collected separately. (2) Color induction. The 20 *N. asiatica* adults were kept in three colored plastic boxes (20 cm × 10 cm × 10 cm, green, yellow, and red) for 24 h, respectively. The other 20 adults were induced without color as a control group. After 24 h, *N. asiatica* adults for the corresponding treatments were collected separately. (3) Insecticide treatment. The *N. asiatica* adults were treated with different concentrations of imidacloprid by spraying method for 24 h. These concentrations included 0.1 mg/L, 0.01 mg/L, and 0.001 mg/L. Twenty individuals were used for each concentration. The control group was 20 *N. asiatica* adults treated with water for 24 h. Then these adults were collected in different 1.5 mL microcentrifuge tubes. (4) Starvation and refeeding treatment. The 20 *N. asiatica* adults were starved for 24 h. The other 20 adults starved for 24 h were refed with 10% honey for 24 h. The normally reared 20 adults were used as the control group. After 24 h, these samples were collected separately. (5) Different temperature treatments. The *N. asiatica* adults were exposed to five different temperatures (including 15 °C, 20 °C, 25 °C, 30 °C, and 35 °C) for 2 h. Twenty individuals were collected for each temperature. (6) Different developmental stages. One hundred eggs (3 days old), 20 3rd instar larvae, 10 pupae (5 days old), and 20 adults (a mixture of males and females, male:female = 1:1) were collected separately. (7) Both sexes. Twenty individuals from each adult female and male *N. asiatica* were collected. (8) Different tissues. Tissues were dissected from *N. asiatica* adults such as 300 antennae, 30 heads without antennae, 30 thorax, 30 abdomen, and 180 legs, kept in different microcentrifuge tubes (1.5 mL, no RNase). All samples collected above were quickly frozen in liquid nitrogen and stored at −80 °C. 

### 4.3. RNA Extraction and cDNA Synthesis

The technique and methods used in this study were similar to our previous studies [33,34]. The RNA and first-strand cDNA of these above-collected samples were extracted and synthesized using the Invitrogen TRIzol Reagent (Invitrogen, Carlsbad, CA, USA) and the primeScriptTM RT reagent Kit with gDNA Eraser (Takara, Dalian, China), respectively. The total RNA quantity and quality were determined by a NanoDrop 2000 spectrophotometer (NanoDrop, Wilmington, DE, USA). Then 1 μg of total RNA was used to synthesize the cDNA. 

### 4.4. Selection and Primer Design of Reference Genes (RGs) in N. asiatica

In this study, a set of 15 genes (*α-TUB*, *β-Actin*, *EF1α*, *EF1β*, *GAPDH*, *G6PDH*, *UBC*, *AK*, *GST*, *SDHA*, *TBP*, *RPL32*, *RPS13*, *RPS15,* and *RPS18*) identified from transcriptome sequencing of *N. asiatica* adults with FPKM value greater than 100, was selected as candidate RGs (Appendix A). These genes had been reported as RGs for qPCR analysis in other insects [20,35,36,37,38,39]. The specific primers of the selected genes were designed using NCBI Primer-BLAST (https://www.ncbi.nlm.nih.gov/tools/primer-blast/index.cgi?LINK_LOC=BlastHome, accessed on 27 November 2021) and listed in Table 1. 

### 4.5. Primer Evaluation and qPCR Analysis

The synthetic cDNA was diluted in a 5-fold series (1/5, 1/25, 1/125, 1/625, and 1/3125), and the dilutions were used to generate a standard curve. The qPCR amplification efficiency (E) (E = (10[−1/slope] − 1) × 100) and correlation coefficient (*R*^2^) of each primer were calculated and analyzed automatically by Bio-Rad CFX 3.0 software. The specificity of qPCR primers was confirmed by the melting curve and sequencing of the qPCR products (Figure 2) [33].

The technique and methods of qPCR used in this study were similar to our previous studies [33,34]. Each qPCR reaction was conducted in a 20 μL reaction: 10 μL of SYBR ^®^ Premix Ex TaqTM (TliRNase H Plus) manual (Takara, China), 0.5 μL of each primer (10 μM), 1 μL of sample cDNA, and 8 μL of sterilized ddH2O. The reactions were performed on a StepOne thermocycler (Bio-Rad CFX, USA) using the two-step method (run as follows: 94 °C for 2 min, followed by 40 cycles of 94 °C for 15 s, 60 °C for 30 s, 60 °C for 1 min) and were analyzed with a melting curve analysis program (heated to 95 °C for 30 s and cooled to 60 °C for 15 s). The Ct values were obtained from qPCR results analyzed by the Bio-Rad CFX 3.0 software. 

### 4.6. Determining the Expression Stability of Candidate Reference Genes

The qPCR data for each of the eight experimental samples were analyzed independently. Each sample for one RG was set as three biological and three technical replicates by qPCR analysis. Four algorithms (∆Ct method [27], GeNorm [28], NormFinder [29], BestKeeper [30]) were used to evaluate the expression stability of 15 candidate RGs. The comprehensive rank was calculated by RefFinder [31], which could unify and merge the above four algorithms. A suitable number of RGs for qPCR normalization was determined by GeNorm analysis [20,28]. Analyses of these algorithms used in this study were similar to previous studies [10,11,12,20,21]. 

### 4.7. Validation of the Selected Reference Genes

To confirm the reliability of the potential reference genes, the relative expression of target *OBP56a* was measured by qPCR at the antennae of both sexes (male and female) and different tissues (antenna, leg, ovipositor) of *N. asiatica* female adults and under odor stimulation (host plant volatiles). The *OBP56a* was identified from transcriptome sequencing of *N. asiatica* adults, whose details (including sequence analysis, and primer evaluation) were listed in Appendix A. The *OBP56a* expression was normalized with the most stable and least stable RGs obtained from the comprehensive analysis of GeNorm and RefFinder. The study method of the *OBP56a* by qPCR was described in detail in Section 4.4. Each experimental sample was set as three biological and three technical replicates in qPCR analysis. The relative expression level of target *OBP56a* was calculated according to the 2^−∆∆Ct^ method [16]. The FPKM value of *OBP56a* in transcriptome data was used as a reference for its expression pattern. 

### 4.8. Statistical Analysis

All statistic comparison was determined using SPSS 22.0 software (SPSS Inc., Chicago, IL, USA). Data multiple comparisons were assessed by ANOVA following Tukey’s HSD multiple comparison test (*p* < 0.05). The statistical significance of the difference between the two treatments was analyzed using a pairwise Student’s *t*-test (*p* < 0.05) [33,34].

## Figures and Tables

**Figure 1 ijms-24-00451-f001:**
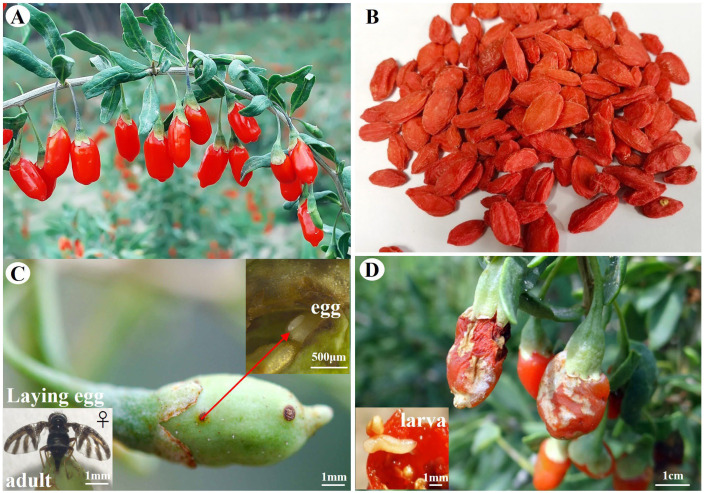
*Lycium barbarum* L. and the symptoms of its fruits harmed by *Neoceratitis asiatica*. (**A**) Mature fruits of *L. barbarum*. (**B**) Dried ripe fruits of *L. barbarum* L. (**C**) Harmful symptoms of *N. asiatica* female adult laying eggs on young fruit (5–7 days after falling flower) of *L. barbarum* L. (**D**) Harmful symptoms of *N. asiatica* larva on mature fruits (28–31 days after falling flower) of *L. barbarum* L.

**Figure 2 ijms-24-00451-f002:**
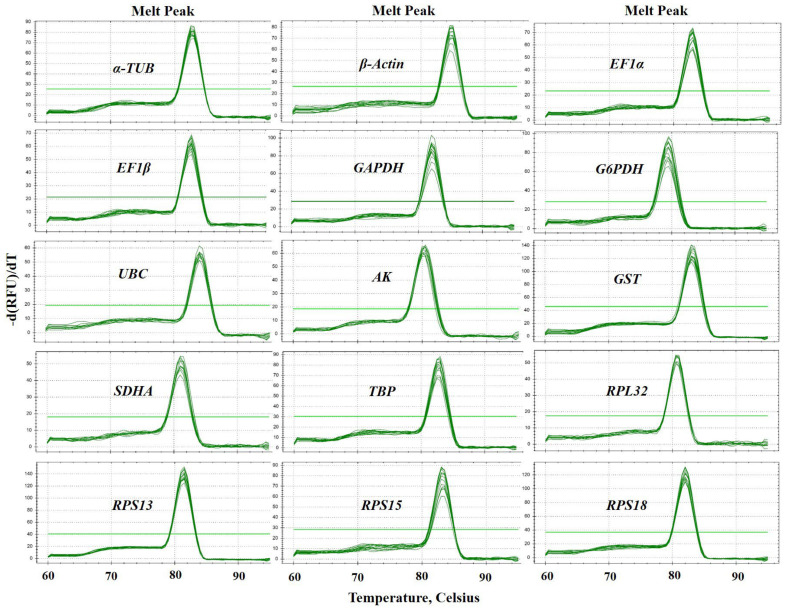
Specificity of primer pairs for qPCR amplification in *N. asiatica*. The melt peaks of primers for qPCR amplification of 15 candidate RGs, including *α-TUB*, *β-Actin*, *EF1α*, *EF1β*, *GAPDH*, *G6PDH*, *UBC*, *AK*, *GST*, *SDHA*, *TBP*, *RPL32*, *RPS13*, *RPS15* and *RPS18*.

**Figure 3 ijms-24-00451-f003:**
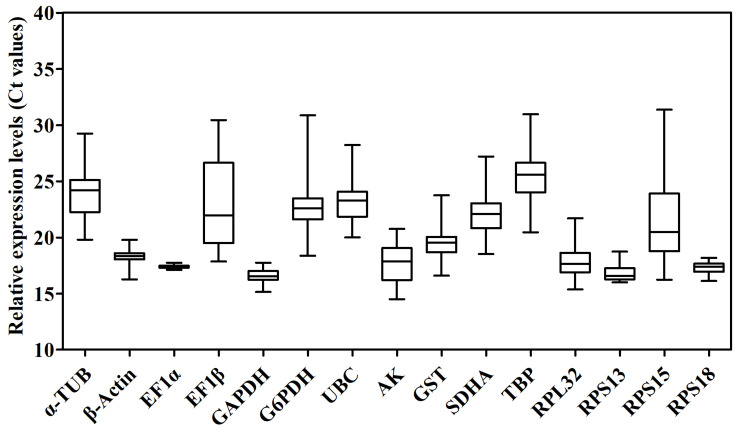
Candidate reference genes expression profiles in *N. asiatica*. The expression data are presented as mean Ct values of candidate reference genes across all samples under eight different experimental conditions. Whiskers represent the maximum and minimum values. The lower and upper borders of the boxes represent the 25th and 75th percentiles, respectively. The line across the box indicates the median Ct value.

**Figure 4 ijms-24-00451-f004:**
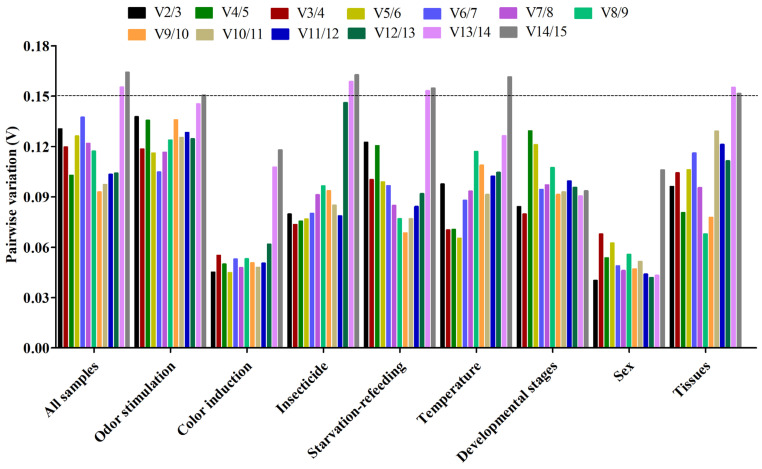
GeNorm analysis of paired variation (V) values of 15 candidate reference genes. Vn/Vn + 1 values are used to determine the optimal number of reference genes. The cut-off value to determine the optimal number of reference genes for qPCR normalization is 0.15.

**Figure 5 ijms-24-00451-f005:**
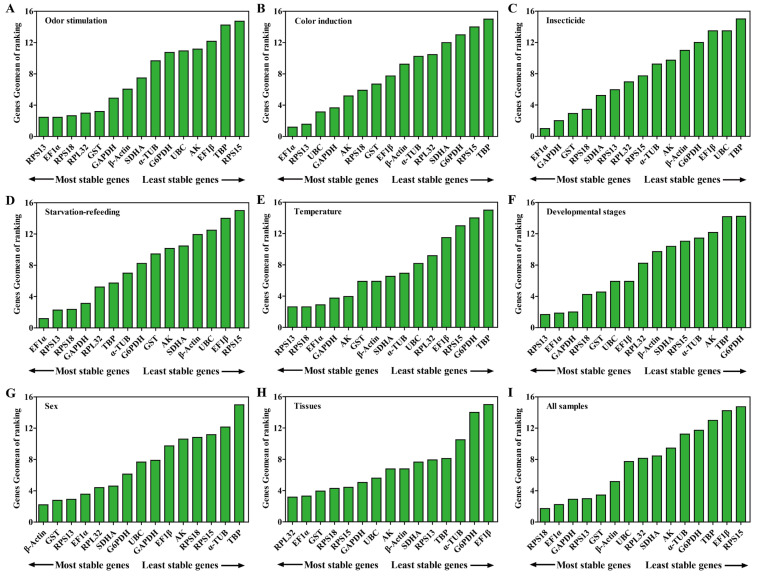
Stability of 15 candidate reference genes in *N. asiatica* under eight experimental conditions by RefFinder. In a RefFinder analysis, increasing Geomean values correspond to decreasing gene expression stability. The Geomean values for the following *N. asiatica* samples are presented: (**A**) Odor stimulation: samples treated with different odor compounds; (**B**) Color induction: samples treated with different colors; (**C**) insecticide treatment: samples treated with imidacloprid insecticide; (**D**) Starvation-refeeding: samples treated with chlorpyrifos; (**E**) Temperature: samples treated with different temperature; (**F**) Developmental stages: samples for all developmental stages; (**G**) Sex: samples for male adults and female adults; (**H**) Tissues: samples for different tissues; (**I**) All samples: all samples for all treatments. The details of 15 candidate RGs are listed in Table 1. The most stable genes are listed on the left, while the least stable genes are listed on the right.

**Figure 6 ijms-24-00451-f006:**
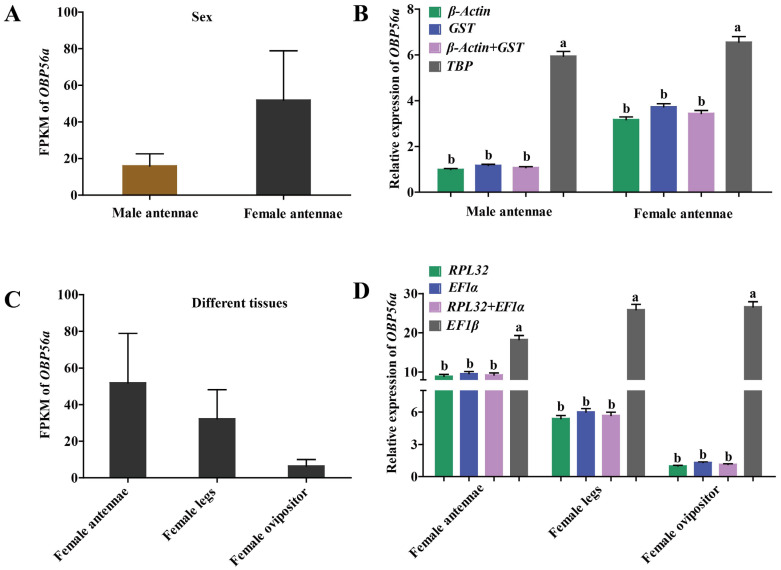
The expression profiles of *OBP56a* in both sexes and in different tissues of *N. asiatica* adults. (**A**) The expression levels of *OBP56a* in the antennae of male and female flies presented by the FPKM value of transcriptome data. FPKM: fragments per kilobase of exon model per million mapped reads. (**B**) The expression levels of *OBP56a* in male and female antennae by qPCR. The expression levels of *OBP56a* in male antennae were used for normalization. Namely, the relative expression levels were presented as fold changes relative to the transcript levels of *OBP56a* in the male antennae. *β-Actin*, *GST*, and *β-Actin* + *GST* were used as the one or two most stable reference genes. *TBP* was used as the least stable reference gene. (**C**) The expression levels of *OBP56a* in different tissues (female antennae, female legs, and female ovipositor) presented by the FPKM value of transcriptome data. (**D**) The expression levels of *OBP56a* in female antennae, female legs, and female ovipositor by qPCR (most stable RGs: *RPL32* and *EF1α*, least stable RG: *EF1β*). The *OBP56a* expression in female ovipositor was used for normalization. Data are represented as the mean ± *SE*. Bars labeled with different letters are significantly different (*p* < 0.05, ANOVA followed by Tukey’s HSD multiple comparison test, n = 3).

**Figure 7 ijms-24-00451-f007:**
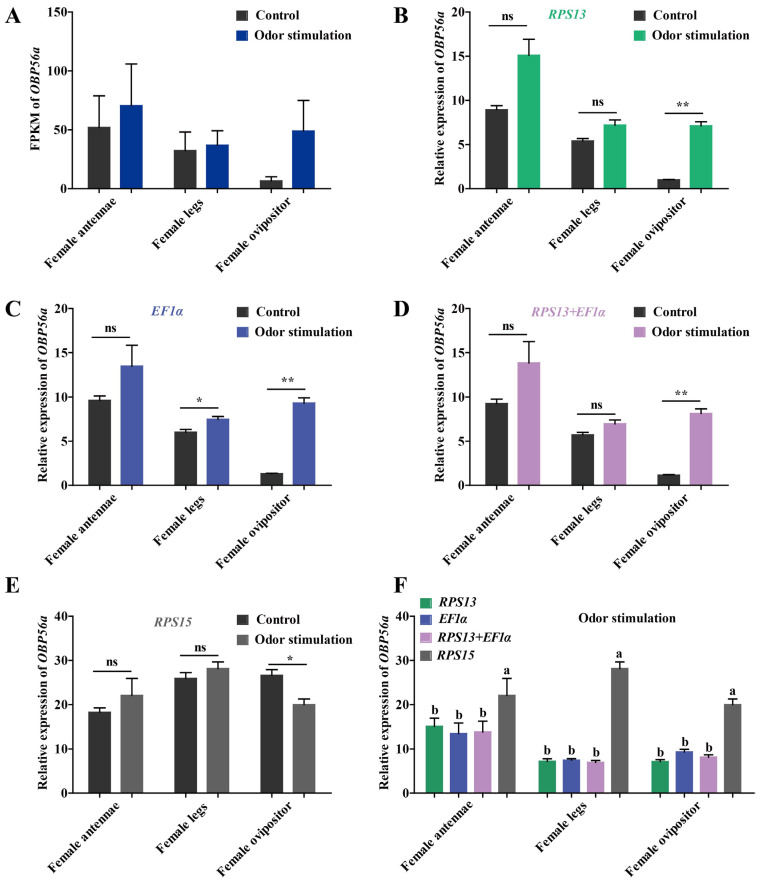
The expression analysis of *OBP56a* in three olfactory organs of female *N. asiatica* adults after odor stimulation. (**A**) The expression levels of *OBP56a* in female antennae, female legs, and female ovipositor before/after odor-stimulation presented by FPKM value of transcriptome data. Control showed the female adults without odor stimulation. Odor sources were collected from the volatiles of wolfberry fruits (green fruits, 5–7 days after falling flower). (**B**–**F**) The expression levels of *OBP56a* in three organs of female adults normalized to the top two stable genes (*RPS13* and *EF1α*) and an unstable gene (*RPS15*) by qPCR. The expression levels of *OBP56a* in female ovipositor were used for normalization. Asterisk (*) indicated a significant difference between control vs. odor stimulation (mean ± *SE*, n = 3, Student’s *t*-test, * *p* < 0.05; ** *p* < 0.01). Bars labeled with different letters are significantly different (mean ± *SE*, n = 3, Tukey’s HSD, *p* < 0.05).

**Table 1 ijms-24-00451-t001:** Primer sequences and amplification sizes of the selected RGs and target gene.

Gene Symbol	Gene Name	Primer Sequence (from 5′ to 3′)	Amplicon Size (bp)	E (%)	*R* ^2^
Reference genes:					
*α-TUB*	Alpha tubulin	F: AAAGGTCATTACACAGAGGGC R: AATGAGCAATGTACCCAGACC	150	95.3	0.998
*β-Actin*	Beta actin	F: GGGATGACATGGAGAAGGTATG R: AGGTCTCGAACATGATTTGGG	144	100.3	0.997
*EF1α*	Elongation factor 1 alpha	F: GCCCAGGTTATTGTGTTGAAC R: GGGATTCTCTTCAGTGGTCTTAC	149	97.9	0.998
*EF1β*	Elongation factor 1 beta	F: TGATGTCTCAACCCCACAAG R: CTGTACGTGTGGGAAGTTACC	148	95.1	0.998
*GAPDH*	Glyceraldehyde-3-phosphate dehydrogenase	F: CCAATGTTCGTTTGCGGTG R: TCTGAGTGGCAGTAGTTGCG	172	101.3	0.997
*G6PDH*	Glucose-6-phosphate dehydrogenase	F: ACTATCTGGGCAAGGAAATGG R: ACACCGAGGCGATATTTTCAC	99	101.4	0.998
*UBC*	Ubiquitin-conjugating enzyme	F: GAAAGATAACTGGGCCGCTG R: GCTACGACTGCATCTTGTGGA	105	100.3	0.996
*AK*	Arginine kinase	F: AGATACAACCTGCAAGTGCG R: ATGCCATCGTACATCTCCTTG	131	109.7	0.996
*GST*	Glutathione S-transferase	F: GCGGCTCATCTATCACTCTAG R: GCTCATGCGTTCTAACCAAAC	122	97.6	0.999
*SDHA*	Succinate dehydrogenase	F: GCAATCAAACCAATACCGCCT R: CATAACGTGTCGGTTCCGGT	198	97.0	0.998
*TBP*	TATA-Box binding protein	F: TACTGGTGCAAAGAGCGAGG R: ACGTCACATGAGCCAACCAT	129	98.0	1.000
*RPL32*	Ribosomal protein L32	F: TCGCATAAATGGCGCAAACC R: AGCATATGACGGGTGCGTTT	116	104.2	0.994
*RPS13*	Ribosomal protein S13	F: CAAGCATTTGGAGCGTAATCG R: AGCGGTACTGGATTCGTATTTC	142	97.2	0.997
*RPS15*	Ribosomal protein S15	F: GGTAAGCGTCAAGTTCTCCTG R: TCGAATTCACCGATGTAGCC	92	95.8	0.997
*RPS18*	Ribosomal protein S18	F: ATCAAAGGTGTGGGTCGC R: TCCTCTTCAGTGCATTCACC	92	96.8	0.998
Target gene:					
*OBP56a*	Odorant binding protein 56a	F: CTGAAACACGCCAAGGAAGC R: CGTTGGGATTGGCGACCTTA	94	103.1	0.994

Note: F: forward primer, R: reverse primer. “E” indicates the primer efficiency for qPCR amplification (E calculated by the standard curve method). “*R*^2^” indicates regression coefficient of the standard curve.

**Table 2 ijms-24-00451-t002:** Expression stability ranking of 15 candidate reference genes in all samples was calculated using ∆Ct, GeNorm and NormFinder. The average expression stability (M value) is listed, and the stability decreased from top to bottom.

Bank	Odor Stimulation		Color Induction		Insecticide Treatment		Starvation- Refeeding		Temperature		Developmental Stages		Sex		Tissues		All Samples	
	Gene	M	Gene	M	Gene	M	Gene	M	Gene	M	Gene	M	Gene	M	Gene	M	Gene	M
**∆Ct analysis**																		
1	*RPS13*	1.24	*EF1α*	0.57	*EF1α*	0.98	*EF1α*	0.94	*RPS13*	1.03	*GAPDH*	0.86	*RPS13*	0.49	*RPL32*	1.60	*RPS18*	1.78
2	*RPL32*	1.26	*UBC*	0.58	*GST*	0.98	*RPS18*	0.95	*GAPDH*	1.09	*RPS13*	0.87	*β-Actin*	0.50	*GST*	1.62	*RPS13*	1.79
3	*GST*	1.29	*RPS13*	0.59	*RPS18*	1.01	*RPS13*	1.00	*AK*	1.10	*EF1α*	0.91	*RPL32*	0.52	*RPS15*	1.63	*GAPDH*	1.79
4	*GAPDH*	1.34	*GAPDH*	0.62	*GAPDH*	1.02	*GAPDH*	1.02	*RPS18*	1.13	*GST*	0.92	*SDHA*	0.52	*UBC*	1.66	*GST*	1.82
5	*RPS18*	1.37	*AK*	0.63	*RPS13*	1.03	*RPL32*	1.03	*SDHA*	1.13	*UBC*	0.99	*GST*	0.54	*TBP*	1.66	*EF1α*	1.83
6	*EF1α*	1.38	*GST*	0.64	*SDHA*	1.04	*TBP*	1.06	*GST*	1.14	*RPS18*	1.00	*EF1α*	0.55	*SDHA*	1.69	*β-Actin*	1.92
7	*SDHA*	1.41	*RPS18*	0.65	*RPS15*	1.14	*α-TUB*	1.09	*EF1α*	1.15	*EF1β*	1.06	*UBC*	0.60	*AK*	1.70	*SDHA*	2.06
8	*β-Actin*	1.53	*EF1β*	0.69	*RPL32*	1.21	*G6PDH*	1.12	*α-TUB*	1.16	*RPL32*	1.09	*G6PDH*	0.61	*GAPDH*	1.73	*UBC*	2.07
9	*α-TUB*	1.76	*β-Actin*	0.71	*α-TUB*	1.28	*GST*	1.15	*β-Actin*	1.16	*β-Actin*	1.09	*EF1β*	0.63	*α-TUB*	1.77	*RPL32*	2.14
10	*UBC*	1.81	*α-TUB*	0.75	*AK*	1.34	*SDHA*	1.17	*UBC*	1.21	*SDHA*	1.20	*GAPDH*	0.64	*EF1α*	1.80	*AK*	2.15
11	*G6PDH*	1.83	*RPL32*	0.77	*β-Actin*	1.40	*AK*	1.25	*EF1β*	1.32	*AK*	1.33	*AK*	0.64	*RPS13*	1.81	*α-TUB*	2.24
12	*AK*	1.97	*SDHA*	0.84	*G6PDH*	1.43	*UBC*	1.41	*RPL32*	1.38	*α-TUB*	1.38	*α-TUB*	0.67	*β-Actin*	1.86	*G6PDH*	2.44
13	*EF1β*	2.11	*G6PDH*	1.00	*EF1β*	2.29	*β-Actin*	1.43	*RPS15*	1.44	*RPS15*	1.39	*RPS15*	0.68	*RPS18*	1.89	*TBP*	2.63
14	*TBP*	3.10	*RPS15*	1.66	*UBC*	2.36	*EF1β*	2.50	*G6PDH*	1.85	*G6PDH*	1.45	*RPS18*	0.76	*G6PDH*	2.37	*EF1β*	3.83
15	*RPS15*	3.45	*TBP*	1.81	*TBP*	3.09	*RPS15*	2.75	*TBP*	2.58	*TBP*	1.58	*TBP*	1.61	*EF1β*	4.08	*RPS15*	4.38
**GeNorm analysis**																		
1	*EF1α*	0.25	*EF1α*	0.14	*EF1α*	0.21	*EF1α*	0.34	*EF1α*	0.24	*EF1α*	0.14	*β-Actin*	0.08	*EF1α*	0.27	*EF1α*	0.53
2	*RPS18*	0.25	*RPS13*	0.14	*GAPDH*	0.21	*RPS13*	0.34	*RPS18*	0.24	*RPS13*	0.14	*GST*	0.13	*RPS18*	0.19	*RPS18*	0.53
3	*RPS13*	0.37	*GAPDH*	0.15	*GST*	0.25	*GAPDH*	0.38	*RPS13*	0.29	*RPS18*	0.22	*EF1α*	0.11	*GAPDH*	0.36	*GAPDH*	0.59
4	*RPL32*	0.43	*UBC*	0.19	*RPS18*	0.28	*RPS18*	0.41	*GAPDH*	0.31	*GAPDH*	0.27	*G6PDH*	0.19	*β-Actin*	0.50	*RPS13*	0.68
5	*GST*	0.54	*RPS18*	0.22	*SDHA*	0.33	*RPL32*	0.51	*β-Actin*	0.34	*EF1β*	0.47	*GAPDH*	0.24	*RPL32*	0.61	*β-Actin*	0.75
6	*GAPDH*	0.62	*AK*	0.25	*RPS13*	0.39	*TBP*	0.56	*α-TUB*	0.38	*GST*	0.58	*RPL32*	0.29	*RPS13*	0.77	*GST*	0.96
7	*β-Actin*	0.67	*GST*	0.29	*RPL32*	0.45	*α-TUB*	0.62	*RPL32*	0.50	*UBC*	0.63	*SDHA*	0.32	*UBC*	1.00	*RPL32*	1.11
8	*SDHA*	0.76	*EF1β*	0.33	*RPS15*	0.53	*G6PDH*	0.66	*UBC*	0.59	*RPL32*	0.69	*RPS13*	0.34	*RPS15*	1.17	*UBC*	1.22
9	*α-TUB*	0.90	*β-Actin*	0.37	*α-TUB*	0.62	*GST*	0.70	*AK*	0.71	*SDHA*	0.77	*UBC*	0.39	*SDHA*	1.28	*SDHA*	1.33
10	*UBC*	1.01	*α-TUB*	0.41	*AK*	0.70	*SDHA*	0.72	*GST*	0.81	*β-Actin*	0.83	*RPS15*	0.42	*AK*	1.35	*AK*	1.41
11	*G6PDH*	1.11	*RPL32*	0.44	*β-Actin*	0.76	*AK*	0.77	*SDHA*	0.86	*α-TUB*	0.89	*EF1β*	0.45	*GST*	1.40	*α-TUB*	1.50
12	*AK*	1.20	*SDHA*	0.48	*G6PDH*	0.81	*UBC*	0.83	*EF1β*	0.94	*RPS15*	0.96	*AK*	0.48	*TBP*	1.4	*G6PDH*	1.60
13	*EF1β*	1.30	*G6PDH*	0.54	*UBC*	1.01	*β-Actin*	0.90	*RPS15*	1.02	*AK*	1.02	*α-TUB*	0.50	*α-TUB*	1.48	*TBP*	1.70
14	*TBP*	1.53	*RPS15*	0.68	*EF1β*	1.19	*EF1β*	1.11	*G6PDH*	1.13	*G6PDH*	1.08	*RPS18*	0.52	*G6PDH*	1.59	*EF1β*	2.01
15	*RPS15*	1.78	*TBP*	0.83	*TBP*	1.44	*RPS15*	1.32	*TBP*	1.33	*TBP*	1.14	*TBP*	0.66	*EF1β*	1.93	*RPS15*	2.33
**NormFinder analysis**																		
1	*RPS13*	0.14	*RPS13*	0.07	*EF1α*	0.11	*RPS18*	0.16	*AK*	0.45	*GAPDH*	0.27	*RPS13*	0.11	*GST*	0.76	*GST*	0.76
2	*RPL32*	0.14	*EF1α*	0.07	*GAPDH*	0.11	*EF1α*	0.16	*GST*	0.52	*RPS13*	0.37	*SDHA*	0.16	*RPS15*	0.82	*RPS13*	0.78
3	*GST*	0.23	*UBC*	0.08	*RPS18*	0.12	*RPS13*	0.31	*SDHA*	0.55	*GST*	0.38	*RPL32*	0.16	*AK*	0.88	*RPS18*	0.82
4	*GAPDH*	0.38	*AK*	0.21	*GST*	0.13	*GAPDH*	0.33	*RPS13*	0.59	*EF1α*	0.50	*β-Actin*	0.28	*RPL32*	0.93	*GAPDH*	0.89
5	*RPS18*	0.53	*GAPDH*	0.23	*SDHA*	0.13	*RPL32*	0.45	*GAPDH*	0.75	*UBC*	0.55	*UBC*	0.30	*UBC*	0.94	*EF1α*	1.02
6	*EF1α*	0.58	*GST*	0.24	*RPS13*	0.23	*TBP*	0.45	*RPS18*	0.80	*RPS18*	0.65	*GST*	0.36	*TBP*	0.96	*β-Actin*	1.10
7	*SDHA*	0.62	*RPS18*	0.28	*RPL32*	0.51	*α-TUB*	0.55	*UBC*	0.82	*EF1β*	0.70	*EF1β*	0.38	*SDHA*	1.02	*UBC*	1.14
8	*β-Actin*	0.88	*EF1β*	0.34	*RPS15*	0.59	*G6PDH*	0.60	*α-TUB*	0.85	*RPL32*	0.73	*AK*	0.39	*α-TUB*	1.11	*AK*	1.28
9	*α-TUB*	1.28	*β-Actin*	0.39	*α-TUB*	0.86	*GST*	0.72	*β-Actin*	0.85	*β-Actin*	0.73	*EF1α*	0.41	*GAPDH*	1.29	*SDHA*	1.31
10	*G6PDH*	1.33	*α-TUB*	0.45	*AK*	0.97	*SDHA*	0.78	*EF1α*	0.88	*SDHA*	0.91	*α-TUB*	0.45	*RPS13*	1.29	*RPL32*	1.47
11	*UBC*	1.37	*RPL32*	0.45	*β-Actin*	1.09	*AK*	0.79	*EF1β*	0.93	*AK*	1.12	*G6PDH*	0.47	*β-Actin*	1.43	*α-TUB*	1.53
12	*AK*	1.49	*SDHA*	0.59	*G6PDH*	1.10	*β-Actin*	1.07	*RPL32*	1.11	*RPS15*	1.16	*RPS15*	0.49	*EF1α*	1.46	*G6PDH*	1.86
13	*EF1β*	1.65	*G6PDH*	0.81	*EF1β*	2.07	*UBC*	1.10	*RPS15*	1.11	*α-TUB*	1.18	*GAPDH*	0.51	*RPS18*	1.59	*TBP*	2.08
14	*TBP*	2.98	*RPS15*	1.60	*UBC*	2.19	*EF1β*	2.39	*G6PDH*	1.66	*G6PDH*	1.28	*RPS18*	0.70	*G6PDH*	2.03	*EF1β*	3.46
15	*RPS15*	3.36	*TBP*	1.76	*TBP*	2.97	*RPS15*	2.67	*TBP*	2.44	*TBP*	1.39	*TBP*	1.59	*EF1β*	3.91	*RPS15*	4.12

**Table 3 ijms-24-00451-t003:** Stability analysis of 15 candidate reference genes based on BestKeeper software. The standard deviation (SD) and coefficient of variation (CV) were given (the lower, the more stable). The stability decreased from top to bottom.

**Bank**	**Odor Stimulation**			**Color Induction**			**Insecticide**			**Starvation-Refeeding**			**Temperature**		
	**Gene**	**SD**	**CV**	**Gene**	**SD**	**CV**	**Gene**	**SD**	**CV**	**Gene**	**SD**	**CV**	**Gene**	**SD**	**CV**
1	*EF1α*	0.01	0.05	*EF1α*	0.01	0.06	*EF1α*	0.10	0.56	*EF1α*	0.11	0.64	*EF1α*	0.02	0.10
2	*RPS18*	0.19	1.09	*RPS13*	0.10	0.59	*GAPDH*	0.11	0.67	*GAPDH*	0.20	1.19	*RPS18*	0.20	1.13
3	*β-Actin*	0.34	1.83	*GAPDH*	0.12	0.78	*GST*	0.14	0.70	*RPS13*	0.23	1.39	*β-Actin*	0.24	1.28
4	*RPS13*	0.39	2.32	*UBC*	0.13	0.62	*RPS18*	0.27	1.53	*RPS18*	0.32	1.84	*RPS13*	0.25	1.49
5	*RPL32*	0.51	2.83	*RPS18*	0.19	1.15	*SDHA*	0.30	1.25	*TBP*	0.37	1.45	*GAPDH*	0.27	1.63
6	*GAPDH*	0.75	4.53	*AK*	0.23	1.54	*RPL32*	0.41	2.25	*RPL32*	0.51	3.04	*α-TUB*	0.33	1.3
7	*GST*	0.78	3.97	*EF1β*	0.27	1.47	*RPS13*	0.41	2.43	*α-TUB*	0.60	2.49	*RPL32*	0.62	3.00
8	*SDHA*	1.08	5.49	*GST*	0.29	1.65	*RPS15*	0.62	3.15	*AK*	0.65	3.54	*UBC*	0.88	3.68
9	*AK*	1.10	5.81	*RPL32*	0.37	2.29	*AK*	0.76	3.88	*G6PDH*	0.67	2.99	*AK*	0.95	5.58
10	*EF1β*	1.22	4.28	*β-Actin*	0.45	2.51	*α-TUB*	0.84	3.21	*β-Actin*	0.77	4.30	*GST*	1.01	4.95
11	*G6PDH*	1.43	6.92	*α-TUB*	0.47	2.14	*β-Actin*	0.89	4.83	*GST*	0.82	4.40	*SDHA*	1.05	4.57
12	*α-TUB*	1.56	7.05	*SDHA*	0.54	2.68	*G6PDH*	0.96	3.86	*SDHA*	0.83	3.71	*EF1β*	1.40	6.61
13	*UBC*	1.70	6.78	*G6PDH*	0.73	3.54	*UBC*	1.61	7.02	*UBC*	0.91	3.82	*RPS15*	1.45	7.40
14	*RPS15*	2.16	7.49	*RPS15*	1.17	5.44	*EF1β*	1.79	7.36	*EF1β*	1.74	6.94	*G6PDH*	1.86	7.82
15	*TBP*	2.79	11.51	*TBP*	1.38	5.68	*TBP*	2.37	9.28	*RPS15*	2.04	7.02	*TBP*	2.27	8.37
**Bank**	**Developmental Stages**			**Sex**			**Tissues**			**All Samples**					
	**Gene**	**SD**	**CV**	**Gene**	**SD**	**CV**	**Gene**	**SD**	**CV**	**Gene**	**SD**	**CV**			
1	*EF1α*	0.02	0.10	*EF1α*	0.02	0.09	*EF1α*	0.04	0.21	*EF1α*	0.11	0.64			
2	*RPS13*	0.13	0.80	*GST*	0.08	0.43	*RPS18*	0.19	1.05	*GAPDH*	0.43	2.63			
3	*RPS18*	0.14	0.87	*β-Actin*	0.12	0.63	*GAPDH*	0.29	1.74	*RPS18*	0.45	2.63			
4	*GAPDH*	0.25	1.50	*RPS18*	0.20	1.15	*β-Actin*	0.46	2.52	*β-Actin*	0.53	2.89			
5	*EF1β*	0.59	3.01	*GAPDH*	0.20	1.23	*RPL32*	0.56	3.25	*RPS13*	0.60	3.59			
6	*GST*	0.71	3.69	*G6PDH*	0.20	0.88	*RPS13*	0.80	4.49	*GST*	1.02	5.26			
7	*UBC*	0.73	3.32	*RPL32*	0.35	1.93	*UBC*	1.31	5.60	*RPL32*	1.16	6.46			
8	*RPS15*	0.73	4.17	*SDHA*	0.36	1.68	*RPS15*	1.44	6.47	*UBC*	1.27	5.48			
9	*RPL32*	0.74	4.26	*RPS13*	0.44	2.60	*SDHA*	1.46	6.3	*SDHA*	1.42	6.46			
10	*α-TUB*	0.83	3.90	*RPS15*	0.46	2.39	*AK*	1.49	8.62	*AK*	1.52	8.65			
11	*β-Actin*	0.93	5.12	*UBC*	0.48	1.98	*GST*	1.55	7.88	*G6PDH*	1.67	7.27			
12	*TBP*	1.02	4.02	*AK*	0.65	3.45	*TBP*	1.60	6.14	*α-TUB*	1.70	7.13			
13	*SDHA*	1.15	5.40	*EF1β*	0.69	3.41	*α-TUB*	1.76	7.13	*TBP*	1.80	7.05			
14	*AK*	1.23	7.05	*α-TUB*	0.78	3.27	*G6PDH*	2.21	9.21	*RPS15*	3.35	15.27			
15	*G6PDH*	1.33	5.88	*TBP*	1.69	6.83	*EF1β*	4.28	16.97	*EF1β*	3.42	14.91			

**Table 4 ijms-24-00451-t004:** Single most stable and optimal combination reference genes in *N. asiatica* under different experimental conditions.

Experimental Conditions	Single Most Stable Reference Genes	Optimal Combination Reference Genes
Odor stimulation	*RPS13*	*RPS13* + *EF1α*
Color induction	*EF1α*	*EF1α* + *RPS13*
Insecticide treatment	*EF1α*	*EF1α* + *GAPDH*
Starvation-refeeding	*EF1α*	*EF1α* + *RPS13*
Different temperature	*RPS13*	*RPS13* + *RPS18*
Developmental stages	*RPS13*	*RPS13* + *EF1α*
Both sexes	*β-Actin*	*β-Actin* + *GST*
Different tissues	*RPL32*	*RPL32* + *EF1α*
All samples	*RPS18*	*RPS18* + *EF1α*

## Data Availability

All data in this study will be available from the corresponding author upon reasonable request.

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
