# Peer review of "Transcriptome-Based Selection and Validation of Reference Genes for Gene Expression in Goji Fruit Fly (Neoceratitis asiatica Becker) under Developmental Stages and Five Abiotic Stresses"

_ijms, 2022, doi:10.3390/ijms24010451_

Round 1

Reviewer 1 Report

The Ms “Transcriptome-based selection and validation of reference  genes for gene expression in Goji fruit fly (Neoceratitis asiatica 3 Becker) under developmental stages and five abiotic stresses” by  wei et al is a nice study. However it needs to be improved before acceptance.

1.       Why 15 genes were selected for analysis? Is their any logic of earlier study behind the selection?

2.       In an earlier study, Actin (ACT), ribosomal protein L32 (rpl32), Glyceraldehyde-3-phosphate dehydrogenase (GAPDH), β-Tubulin, α-Tubulin, and Succinate dehydrogenase (SDH) genes were analysed as reference gene in Phenacoccus solenopsis (https://www.nature.com/articles/s41598-017-13925-9). Similarly a few other genes have been studied in https://www.researchgate.net/journal/Frontiers-in-Physiology-1664-042X  Do you have seen the expression of these genes in your study? How your study is significant that the earlier such reports? Because several earlier studies have reported numerous other genes as reference gene. All those studies should be comparatively discussed in the discussion section.

3.       I could see large variation in the amplicon size from 90-~200 bp. Why the amplicon was not selected for the same length?

4.       Fig 2 should be improved.

5.       Fig 3, I could see high variation in the cT value. Why?

6.       Tables needs to be rearranged.

7.       Why there is no std err bar in fig 5?

8.       I could see numerous typos and grammatical errors that needs to be rectified.

9.       Discussion should be focussed on the results, but should not be repetition of results.

10.   Conclusion needs to rewritten with future perspectives. 

Author Response

Thanks again for your comments! We improved this manuscript according your suggestions. Please see the attachment.

Reviewer 2 Report

This manuscript "Transcriptome-based selection and validation of reference genes for gene expression in Goji fruit fly (Neoceratitis asiatica Becker) under developmental stages and five abiotic stresses" is well written. However, there are few points author should consider before it published.

Results

Line 94: N. asiatica should make italic and please check in whole manuscript.

Table 2 and 3.  Please make it clear. Even it is big and many information, you can arrange in landscape and with small fonts.

Write all gene name in italic font

Figure 4. caption: paired variation (V) values……

Author Response

(The authors gave the same response as above.)

Round 2

Reviewer 1 Report

Authors have revised the manuscript sincerely. It may now be accepted for publication.